# Many Paths to Equilibrium: GANs Do Not Need to Decrease a Divergence At Every Step

William Fedus[* 1], Mihaela Rosca[* 2], Balaji Lakshminarayanan[2],
Andrew M. Dai[1], Shakir Mohamed[2] and Ian Goodfellow[1]

[1]Google Brain
[2]DeepMind

## Abstract

Generative adversarial networks (GANs) are a family of generative models that do not minimize a single training criterion. Unlike other generative models, the data distribution is learned via a game between a generator (the generative model) and a discriminator (a teacher providing training signal) that each minimize their own cost. GANs are designed to reach a Nash equilibrium at which each player cannot reduce their cost without changing the other players' parameters. One useful approach for the theory of GANs is to show that a divergence between the training distribution and the model distribution obtains its minimum value at equilibrium. Several recent research directions have been motivated by the idea that this divergence is the primary guide for the learning process and that every step of learning should decrease the divergence. We show that this view is overly restrictive. During GAN training, the discriminator provides learning signal in situations where the gradients of the divergences between distributions would not be useful. We provide empirical counterexamples to the view of GAN training as divergence minimization. Specifically, we demonstrate that GANs are able to learn distributions in situations where the divergence minimization point of view predicts they would fail. We also show that gradient penalties motivated from the divergence minimization perspective are equally helpful when applied in other contexts in which the divergence minimization perspective does not predict they would be helpful. This contributes to a growing body of evidence that GAN training may be more usefully viewed as approaching Nash equilibria via trajectories that do not necessarily minimize a specific divergence at each step.

## 1 Introduction

Generative adversarial networks (GANs) (Goodfellow et al., 2014) are generative models based on a competition between a generator network $G$ and a discriminator network $D$. The generator network $G$ represents a probability distribution $p_{\text{model}}(\boldsymbol{x})$. To obtain a sample from this distribution, we apply the generator network to a noise vector $\boldsymbol{z}$ sampled from $p_{\boldsymbol{z}}$, that is $\boldsymbol{x} = G(\boldsymbol{z})$. Typically, $\boldsymbol{z}$ is drawn from a Gaussian or uniform distribution, but any distribution with sufficient diversity is possible. The discriminator $D(\boldsymbol{x})$ attempts to distinguish whether an input value $\boldsymbol{x}$ is real (came from the training data) or fake (came from the generator).

The goal of the training process is to recover the true distribution $p_{\text{data}}$ that generated the data. Several variants of the GAN training process have been proposed. Different variants of GANs have been interpreted as approximately minimizing different divergences or distances between $p_{\text{data}}$ and $p_{\text{model}}$. However, it has been difficult to understand whether the improvements are caused by a change in the underlying divergence or the learning dynamics.

We conduct several experiments to assess whether the improvements associated with new GAN methods are due to the reasons cited in their design motivation. We perform a comprehensive study of GANs on simplified, synthetic tasks for which the true $p_{\text{data}}$ is known and the relevant distances

---

[*] Equal contribution.

are straightforward to calculate, to assess the performance of proposed models against baseline methods. We also evaluate GANs using several independent evaluation measures on real data to better understand new approaches. Our contributions are:

- We aim to clarify terminology used in recent papers, where the terms "standard GAN," "regular GAN," or "traditional GAN" are used without definition (e.g., (Arjovsky et al., 2017; Denton et al., 2015; Salimans et al., 2016; Donahue et al., 2016)). The original GAN paper described two different losses: the "minimax" loss and the "non-saturating" loss, equations (10) and (13) of Goodfellow (2016), respectively. Recently, it has become important to clarify this terminology, because many of the criticisms of "standard GANs", e.g. Arjovsky et al. (2017), are applicable only to the minimax GAN, while the non-saturating GAN is the standard for GAN implementations. The non-saturating GAN was recommended for use in practice and implemented in the original paper of Goodfellow et al. (2014), and is the default in subsequent papers (Radford et al., 2015; Salimans et al., 2016; Donahue et al., 2016; Nowozin et al., 2016)[1]. To avoid confusion we will always indicate whether we mean minimax GAN (M-GAN) or non-saturating GAN (NS-GAN).

- We demonstrate that gradient penalties designed in the divergence minimization framework—to improve Wasserstein GANs (Gulrajani et al., 2017) or justified from a game theory perspective to improve minimax GANs (Kodali et al., 2017)—also improve the non-saturating GAN on both synthetic and real data. We observe improved sample quality and diversity.

- We find that non-saturating GANs are able to fit problems that cannot be fit by Jensen-Shannon divergence minimization. Specifically, Figure 1 shows a GAN using the loss from the original non-saturating GAN succeeding on a task where the Jensen-Shannon divergence provides no useful gradient. Figure 2 shows that the non-saturating GAN does not suffer from vanishing gradients when applied to two widely separated Gaussian distributions.

## 2 VARIANTS OF GENERATIVE ADVERSARIAL NETWORKS

### 2.1 NON-SATURATING AND MINIMAX GANS

In the original GAN formulation (Goodfellow et al., 2014), the output of the discriminator is a probability and the cost function for the discriminator is given by the negative log-likelihood of the binary discrimination task of classifying samples as real or fake:

$$J^{(D)}(D, G) = - \mathop{\mathbb{E}}_{\boldsymbol{x} \sim p_{\text{data}}} \left[ \log D(\boldsymbol{x}) \right] - \mathop{\mathbb{E}}_{\boldsymbol{z} \sim p_{\boldsymbol{z}}} \left[ \log(1 - D(G(\boldsymbol{z}))) \right]. \tag{1}$$

The theoretical analysis in (Goodfellow et al., 2014) is based on a zero-sum game in which the generator maximizes $J^{(D)}$, a situation that we refer to here as "minimax GANs". In minimax GANs the generator attempts to generate samples that have low probability of being fake, by minimizing the objective (2). However, in practice, Goodfellow et al. (2014) recommend implementing an alternative cost function that instead ensures that generated samples have high probability of being real, and the generator instead minimizes an alternative objective (3).

$$\textbf{Minimax} \quad J^{(G)}(G) = \mathop{\mathbb{E}}_{\boldsymbol{z} \sim p_{\boldsymbol{z}}} \log[1 - D(G(\boldsymbol{z}))]. \tag{2}$$

$$\textbf{Non-saturating} \quad J^{(G)}(G) = - \mathop{\mathbb{E}}_{\boldsymbol{z} \sim p_{\boldsymbol{z}}} \log D(G(\boldsymbol{z})). \tag{3}$$

We refer to the alternative objective as non-saturating, due to the non-saturating behavior of the gradient (see figure 2), and was the implementation used in the code of the original paper. We use the non-saturating objective (3) in all our experiments

---

[1] The original GAN paper implements both the minimax and non-saturating cost but uses the non-saturating cost for the published configurations of experiments: https://github.com/goodfeli/adversarial/blob/master/cifar10_convolutional.yaml#L139. To the best of our knowledge, the DCGAN codebase implements *only* the non-saturating cost: https://github.com/soumith/dcgan.torch/blob/master/main.lua#L215. Likewise, the improved-gan codebase implements *only* the non-saturating cost: https://github.com/openai/improved-gan/blob/master/imagenet/build_model.py#L114. If only one of these two costs were to be called "standard," it should be the non-saturating version.

As shown in (Goodfellow et al., 2014), whenever $D$ successfully minimizes $J^{(D)}$ optimally, maximizing $J^{(D)}$ with respect to the generator is equivalent to minimizing the Jensen-Shannon divergence. Goodfellow et al. (2014) use this observation to establish that there is a unique Nash equilibrium in function space corresponding to $p_{\text{data}} = p_{\text{model}}$.

## 2.2 WASSERSTEIN GAN

Wasserstein GANs (Arjovsky et al., 2017) modify the discriminator to emit an unconstrained real number rather than a probability (analogous to emitting the logits rather than the probabilities used in the original GAN paper). The cost function for the WGAN then omits the log-sigmoid functions used in the original GAN paper. The cost function for the discriminator is now:

$$W^{(D)}(D, G) = \underset{\boldsymbol{x} \sim p_{\text{data}}}{\mathbb{E}} [D(\boldsymbol{x})] - \underset{\boldsymbol{z} \sim p_{\boldsymbol{z}}}{\mathbb{E}} [D(G(\boldsymbol{z}))]. \tag{4}$$

The cost function for the generator is simply $W^{(G)} = -W^{(D)}(D, G)$. When the discriminator is Lipschitz smooth, this approach approximately minimizes the earth mover's distance between $p_{\text{data}}$ and $p_{\text{model}}$. To enforce Lipschitz smoothness, the weights of $D$ are clipped to lie within $(-c, c)$ where $c$ is some small real number.

## 2.3 GRADIENT PENALTIES FOR GENERATIVE ADVERSARIAL NETWORKS

Multiple formulations of gradient penalties have been proposed for GANs. As introduced in WGAN-GP (Gulrajani et al., 2017), the gradient penalty is justified from the perspective of the Wasserstein distance, by imposing properties which hold for an optimal critic as an additional training criterion. In this approach, the gradient penalty is typically a penalty on the gradient norm, and is applied on a linear interpolation between data points and samples, thus smoothing out the space between the two distributions.

Kodali et al. (2017) introduce DRAGAN with a gradient penalty from the perspective of regret minimization, by setting the regularization function to be a gradient penalty on points around the data manifold, as in *Follow The Regularized Leader* (Cesa-Bianchi & Lugosi, 2006), a standard no-regret algorithm. This encourages the discriminator to be close to linear around the data manifold, thus bringing the set of possible discriminators closer to a convex set, the set of linear functions. We also note that they used the minimax version of the game to define the loss, in which the generator maximizes $J^{(D)}$ rather than minimizing $J^{(G)}$.

To formalize the above, both proposed gradient penalties of the form:

$$\underset{\hat{x} \sim p_{\hat{x}}}{\mathbb{E}} \left[ (\|\nabla_{\hat{x}} D(\hat{x})\|_2 - 1)^2 \right], \tag{5}$$

where $p_{\hat{x}}$ is defined as the distribution defined by the sampling process:

$$x \sim p_{\text{data}}; \qquad x_{\text{model}} \sim p_{\text{model}}; \qquad x_{\text{noise}} \sim p_{\text{noise}} \tag{6}$$

$$\textbf{DRAGAN} \quad \tilde{x} = x + x_{\text{noise}} \tag{7a}$$

$$\textbf{WGAN-GP} \quad \tilde{x} = x_{\text{model}} \tag{7b}$$

$$\alpha \sim U(0, 1) \tag{8}$$

$$\hat{x} = \alpha x + (1 - \alpha)\tilde{x}. \tag{9}$$

As we will note in our experimental section, Kodali et al. (2017) also reported that mode-collapse is reduced using their version of the gradient penalty.

### 2.3.1 NON-SATURATING GAN WITH GRADIENT PENALTY

We consider the non-saturating GAN objective (3) supplemented by two gradient penalties: the penalty proposed by Gulrajani et al. (2017), which we refer to as "GAN-GP"; the gradient penalty

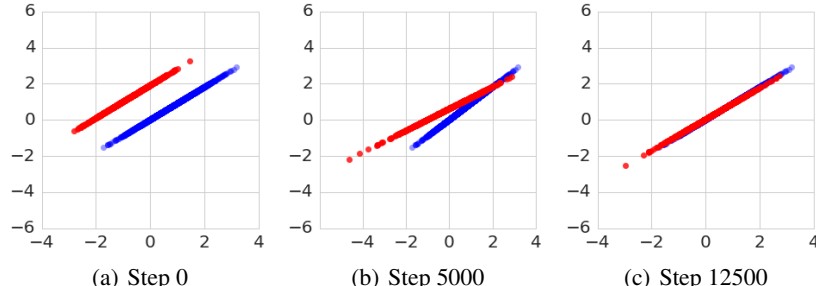

(a) Step 0      (b) Step 5000      (c) Step 12500

Figure 1: Visualization of experiment 1 training dynamics in two dimensions, demonstrated specifically in the case where the model is initialized so that it it represents a linear manifold parallel to the linear manifold of the training data. Here the GAN model (red points) converges upon the one dimensional synthetic data distribution (blue points). Specifically, this is an illustration of the parallel line thought experiment from (Arjovsky et al., 2017). When run in practice with a non-saturating GAN, the GAN succeeds. In the same setting, minimization of Jensen-Shannon divergence would fail. This indicates that while Jensen-Shannon divergence is useful for characterizing GAN equilibrium, it does not necessarily tell us much about non-equilibrium learning dynamics.

proposed by DRAGAN (Kodali et al., 2017), which we refer to as DRAGAN-NS, to emphasize that we use the non-saturating generator loss function. In both cases, the gradient penalty applies only to the discriminator, with the generator loss remaining unchanged (as defined in Equation 3). In this setting, the loss of the discriminator becomes:

$$\tilde{J}^{(D)}(D, G) = - \underset{x \sim p_{\text{data}}}{\mathbb{E}} \left[\log D(x)\right] - \underset{z \sim p_z}{\mathbb{E}} \left[\log(1 - D(G(z)))\right] + \lambda \underset{\hat{x} \sim p_{\hat{x}}}{\mathbb{E}} \left[(\|\nabla_{\hat{x}} D(\hat{x})\|_2 - 1)^2\right]$$

(10)

We consider these GAN variants because:

- We want to assess whether gradient penalties are effective outside their original defining scope. Namely, we perform experiments to determine whether the benefit obtained by applying the gradient penalty for Wasserstein GANs is obtained from properties of the earth mover's distance, or from the penalty itself. Similarly, we evaluate whether the DRAGAN gradient penalty is beneficial outside the minimax GAN setting.

- We want to assess whether the exact form of the gradient penalty matters.

- We compare three models, to control over different aspects of training: same gradient penalty but different underlying adversarial losses (GAN-GP versus WGAN-GP), as well as the same underlying adversarial loss, but different gradient penalties (GAN-GP versus DRAGAN-NS).

We note that we do not compare with the original DRAGAN formulation, which uses the minimax GAN formulation, since in this work we focus on non-saturating GAN variants.

## 3   MANY PATHS TO EQUILIBRIUM

The original GAN paper (Goodfellow et al., 2014) used the correspondence between $J^{(D)}(D^*, G)$ and the Jensen-Shannon divergence to characterize the *Nash equilibrium* of minimax GANs. It is important to keep in mind that there are many ways for the learning process to approach this equilibrium point, and the majority of them do not correspond to gradually reducing the Jensen-Shannon divergence at each step. Divergence minimization is useful for understanding the outcome of training, but GAN training is not the same thing as running gradient descent on a divergence and GAN training may not encounter the same problems as gradient descent applied to a divergence.

Arjovsky et al. (2017) describe the *learning process* of GANs from the perspective of divergence minimization and show that the Jensen-Shannon divergence is unable to provide a gradient that will bring $p_{\text{data}}$ and $p_{\text{model}}$ together if both are sharp manifolds that do not overlap early in the learning

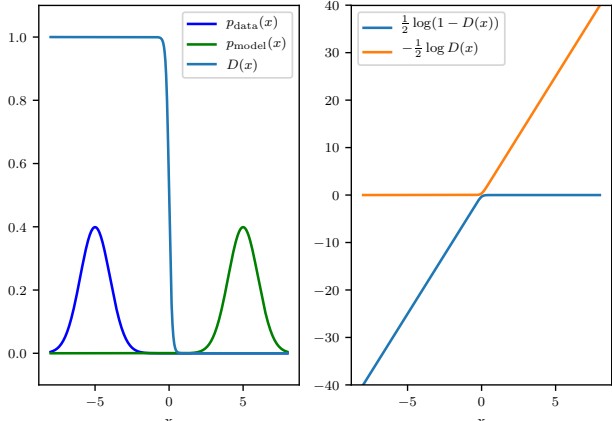

Figure 2: (Left) A recreation of Figure 2 of Arjovsky et al. (2017). This figure is used by Arjovsky et al. (2017) to show that a model they call the "traditional GAN" suffers from vanishing gradients in the areas where $D(x)$ is flat. This plot is correct if "traditional GAN" is used to refer to the *minimax* GAN, but it does not apply to the *non-saturating* GAN. (Right) A plot of both generator losses from the original GAN paper, as a function of the generator output. Even when the model distribution is highly separated from the data distribution, non-saturating GANs are able to bring the model distribution closer to the data distribution because the *loss function* has strong gradient when the generator samples are far from the data samples, *even when the discriminator itself has nearly zero gradient*. While it is true that the $\frac{1}{2} \log(1 - D(x))$ loss has a vanishing gradient on the right half of the plot, the original GAN paper instead recommends implementing $-\frac{1}{2} \log D(x)$. This latter, recommended loss function has a vanishing gradient only on the left side of the plot. It makes sense for the gradient to vanish on the left because generator samples in that area have already reached the area where data samples lie.

process. Following this line of reasoning, they suggest that when applied to probability distributions that are supported only on low dimensional manifolds, the Kullback Leibler (KL), Jensen Shannon (JS) and Total Variation (TV) divergences do not provide a useful gradient for learning algorithms based on gradient descent, the "traditional GANs" is inappropriate for fitting such low dimensional manifolds ("traditional GAN" seems to refer to the minimax version of GANs used for theoretical analysis in the original paper, and there is no explicit statement about whether the argument is intended to apply to the non-saturating GAN implemented in the code accompanying the original GAN paper). In Section 4 we show that non-saturating GANs are able to learn on tasks where the data distribution lies on the low dimensional manifold.

We show that non-saturating GANs do not suffer from vanishing gradients for two widely separated Gaussians in Figure 2. The fact that the gradient of the recommended loss does not actually vanish explains why GANs with the non-saturating objective (3), are able to bring together two widely separated Gaussian distributions. Note that the gradient for this loss does not vanish *even when the discriminator is optimal*. The *discriminator* has vanishing gradients but the *generator loss* amplifies small differences in discriminator outputs to recover strong gradients. This means it is possible to train the GAN by changing the *loss* rather than the discriminator.

For the parallel lines thought experiment (Arjovsky et al., 2017) (see Figure 1), the main problem with the Jensen-Shannon divergence is that it is parameterized in terms of the density function, and the two density functions have no support in common. Most GANs, and many other models, can solve this problem by parameterizing their loss functions in terms of samples from the two distributions rather than in terms of their density functions.

## 4 SYNTHETIC EXPERIMENTS

To assess the learning process of GANs we empirically examine GAN training on pathological tasks where the data is constructed to lie on a low dimensional manifold, and show the model is able to learn the data distribution in cases where using the underlying divergence obtained at optimality would not provide useful gradients. We then evaluate convergence properties of common GAN variants on this task where the parameters generating the distribution are known.

### 4.1 EXPERIMENT I: 1-D DATA MANIFOLD AND 1-D GENERATOR

In our first experiment, we generate synthetic training data that lies along a one-dimensional line and design a one-dimensional generative model, however, we embed the problem in a higher $d$-dimensional space where $d \gg 1$. This experiment is essentially an implementation of a thought experiment from Arjovsky et al. (2017).

Specifically, in a $d$-dimensional space, we define $p_{\text{data}}$ by randomly generating parameters defining the distribution once at the beginning of the experiment. We generate a random $b_r \in \mathbb{R}^d$ and random $W_r \in \mathbb{R}^{1 \times d}$. Our latent $z_r \in \mathbb{R} \sim N(0, \sigma)$ where $\sigma$ is the standard deviation of the normal distribution. The synthetic training data of $m$ examples is then given by

$$\{x^{(i)}\}_{i=1}^m = \{z_r^{(i)}\}_{i=1}^m W_r + b_r \tag{11}$$

The real synthetic data is therefore Gaussian distributed on a 1-D surface within the space, where the position is determined by $b_r$ and the orientation is determined by $W_r$.

The generator also assumes the same functional form, that is, it is also intrinsically one dimensional,

$$G_\theta(z) = zW_\theta + b_\theta \tag{12}$$

where $b_\theta \in \mathbb{R}^d$ and $W_\theta \in \mathbb{R}^{1 \times d}$. The discriminator is a single hidden layer ReLU network, which is of higher complexity than the generator so that it may learn non-linear boundaries in the space.

This experiment captures the idea of sharp, non-overlapping manifolds that motivate alternative GAN losses. Further, because we know the true generating parameters of the training data, we may explicitly test convergence properties of the various methodologies.

### 4.2 EXPERIMENT II: 1-D DATA MANIFOLD AND OVERCOMPLETE GENERATOR

In our second experiment, the synthetic training data is still the same (lying on a 1-D line) and given by Eq. 11 but now the generator is overcomplete for this task, and has a higher latent dimension $g$, where $1 < g \le d$.

$$G(z) = zW_\theta + b_\theta \tag{13}$$

where matrix $W_\theta \in \mathbb{R}^{g \times d}$ and vector $b_\theta \in \mathbb{R}^d$, so that the generator is able to represent a manifold with too high of a dimensionality. The generator parameterizes a multivariate Gaussian $N(x; \mu, \Sigma)$ with $\mu = b$. The covariance matrix elements $\Sigma_{ij} = E[\sigma^2(X_i - \mu_i)(X_j - \mu_j)] = \sigma^2 E[(X_i - \mu_i)(X_j - \mu_j)]$. In vector notation, $\Sigma = \sigma^2 W^T W$.

### 4.3 RESULTS

To evaluate the convergence of an experimental trial, we report the square Fréchet distance (Fréchet (1957)) between the true data Gaussian distribution and the fitted Gaussian parameters. In our notation, where the $r$ subscript denotes real data, the $\theta$ subscript denotes the generator Gaussian parameters and $\|x\|^2$ is the squared $l_2$ norm of $x$, the Fréchet distance is defined as (Dowson & Landau (1982)):

$$d^2(\mu_r, \mu_\theta, \Sigma_r, \Sigma_\theta) = \|\mu_\theta - \mu_r\|^2 + \text{Tr}\left(\Sigma_r + \Sigma_\theta - 2(\Sigma_r \Sigma_\theta)^{-1/2}\right) \tag{14}$$

Every GAN variant was trained for 200000 steps. For each step, the generator is updated once and the discriminator is updated 5 times. Throughout the paper, the number of steps will correspond to the number of generator updates.

The main conclusions from our synthetic data experiments are:

- Gradient penalties (both applied near the data manifold, DRAGAN-NS, and at an interpolation between data and samples, GAN-GP) stabilize training and improve convergence (Figures 3, 9, 10).

- Despite the inability of Jensen-Shannon divergence minimization to solve this problem, we find that the non-saturating GAN succeeds in converging to the 1D data manifold (Figure 3). However, in higher dimensions the resulting fit is not as strong as the other methods: Figure 10 shows that increasing the number of dimensions while keeping the learning rate fixed can decrease the performance of the non-saturating GAN model.

- Non-saturating GANs are able to learn data distributions which are disjoint from the training sample distribution at initialization (or another point in training), as demonstrated in Figure 1.

- Updating the discriminator 5 times per generator update does not result in vanishing gradients when using the non-saturating cost. However, when scaling the number of discriminator updates to 100 per generator update, non-saturating GANs perform worse than when using a smaller number of updates (1, 5, 10). Gradient penalties help here too: GAN-GP scales better with the number of discriminator updates. The results are detailed in Appendix Section A.2.

- An over-capacity generator with the ability to have more directions of high variance than the underlying data is able to capture the data distribution using non-saturating GAN training (Figure 4).

## 4.4 HYPERPARAMETER SENSITIVITY

We assess the robustness of the considered models by looking at results across hyperparameters for both experiment 1 and experiment 2. In one setting, we keep the input dimension fixed while varying the learning rate (Figure 9); in another setting, we keep the learning rate fixed, while varying the input dimension (Figure 10). In both cases, the results are averaged out over 1000 runs per setting, each starting from a different random seed. We notice that:

- The non-saturating GAN model (with no gradient penalty) is most sensitive to hyperparameters.

- Gradient penalties make the non-saturating GAN model more robust.

- Both Wasserstein GAN formulations are quite robust to hyperparameter changes.

- For certain hyperparameter settings, there is no performance difference between the two gradient penalties for the non-saturating GAN, when averaging across random seeds. This is especially visible in Experiment 1, when the number of latent variables is 1. This could be due to the fact that the data sits on a low dimensional manifold, and because the discriminator is a small, shallow network.

## 5 REAL DATA EXPERIMENTS

To assess the effectiveness of the gradient penalty on standard datasets for the non-saturating GAN formulation, we train a non-saturating GAN, a non-saturating GAN with the gradient penalty introduced by (Gulrajani et al., 2017) (denoted by GAN-GP), a non-saturating GAN with the gradient penalty introduced by (Kodali et al., 2017) (denoted by DRAGAN-NS), and a Wasserstein GAN with gradient penalty (WGAN-GP) on three datasets: Color MNIST (Metz et al., 2016) - data dimensionality $(28, 28, 3)$, CelebA (Liu et al., 2015) - data dimensionality $(64, 64, 3)$ and CIFAR-10 (Krizhevsky, 2009) - data dimensionality $(32, 32, 3)$, as seen in Figure 6.

For all our experiments we used $\lambda = 10$ as the gradient penalty coefficient and used batch normalization (Ioffe & Szegedy, 2015); Kodali et al. (2017) suggests that batch normalization is not neeeded

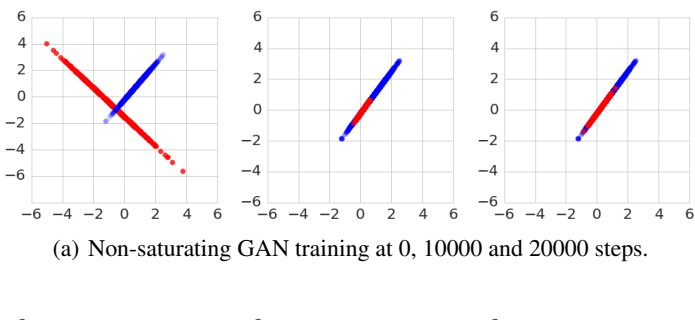

(a) Non-saturating GAN training at 0, 10000 and 20000 steps.

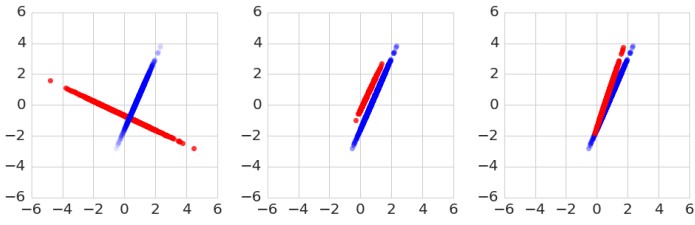

(b) GAN-GP training at 0, 10000 and 20000 steps.

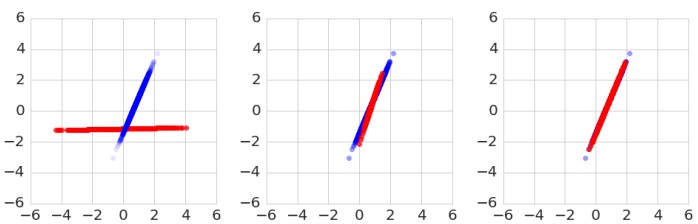

(c) DRAGAN-NS training at 0, 10000 and 20000 steps.

Figure 3: Visualization of experiment 1 training dynamics in two dimensions. Here the GAN model (red points) converges upon the one dimensional synthetic data distribution (blue points). We note that this is a visual illustration, and the results have not been averaged out over multiple seeds. Exact plots may vary on different runs. However, a single example of success is sufficient to refute claims that this this task is impossible for this model.

for DRAGAN, but we found that it also improved our DRAGAN-NS results. We used the Adam optimizer (Kingma & Ba, 2014) with $\beta_1 = 0.5$ and $\beta_2 = 0.9$ and a batch size of 64. The input data was scaled to be between -1 and 1. We did not add any noise to the discriminator inputs or activations, as that regularization technique can be interpreted as having the same goal as gradient penalties, and we wanted to avoid a confounding factor. We trained all Color MNIST models for 100000 steps, and CelebA and CIFAR-10 models for 200000 steps. We note that the experimental results on real data for the non-saturating GAN and for the Improved Wasserstein GAN (WGAN-GP) are quoted with permission from an earlier publication by Rosca et al. (2017).

We note that the WGAN-GP model was the only model for which we did 5 discriminator updates in real data experiments. All other models (DCGAN, DRAGAN-NS, GAN-GP) used one discriminator update for generator update.

For all reported results, we sweep over two hyperparameters:

- Learning rates for the discriminator and generator. Following Radford et al. (2015), we tried learning rates of 0.0001, 0.0002, 0.0003 for both the discriminator and the generator. We note that this is consistent with WGAN-GP, where the authors use 0.0002 for CIFAR-10 experiments.

- Number of latents. For CelebA and CIFAR-10 we try latent sizes 100, 128 and 150, while for Color MNIST we try 10, 50, 75.

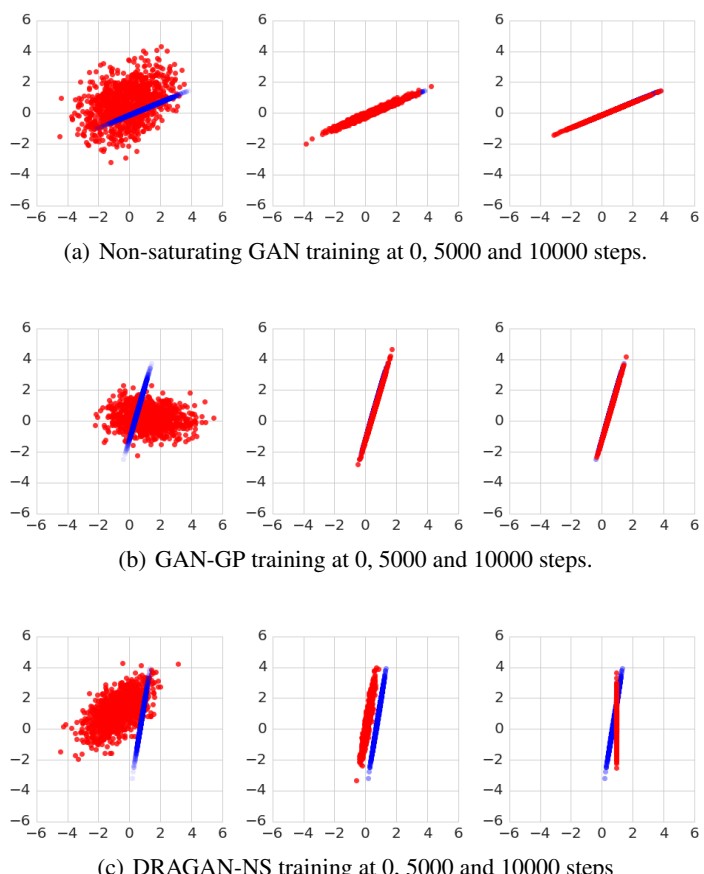

(a) Non-saturating GAN training at 0, 5000 and 10000 steps.

(b) GAN-GP training at 0, 5000 and 10000 steps.

(c) DRAGAN-NS training at 0, 5000 and 10000 steps

Figure 4: Visualization of experiment 2 training dynamics in two dimensions - where the GAN model has 3 latent variables. Here the rank one GAN model (red points) converges upon the one dimensional synthetic data distribution (blue points). We observe how for poor initialization the non-saturating GAN suffers from mode collapse. However, adding a gradient penalty stabilizes training. We note that this is a visual illustration, and the results have not been averaged out over multiple seeds. Exact plots may vary on different runs.

## 5.1 EVALUATION

Unlike the synthetic case, here we are unable to evaluate the performance of our models relative to the true solution, since that is unknown. Moreover, there is no single metric that can evaluate the performance of GANs. We thus complement visual inspection with three metrics, each measuring a different criteria related to model performance. We use the Inception Score (Salimans et al., 2016) to measure how visually appealing CIFAR-10 samples are, the MS-SSIM metric (Wang et al., 2003; Odena et al., 2016) to check sample diversity, and an Improved Wasserstein independent critic to assess overfitting, as well as sample quality (Danihelka et al., 2017). For a more detailed explanation of these metrics, we refer to Rosca et al. (2017). In all our experiments, we control over discriminator and generator architectures, using the ones used by DCGAN (Radford et al., 2015) and the original WGAN paper (Arjovsky et al., 2017)[2]. We note that the WGAN-GP paper used a different architecture when reporting the Inception Score on CIFAR10, and thus their results are not directly comparable.

For all the metrics, we report both the hyperparameter sensitivity of the model (by showing quartile statistics), as well as the 10 best results according to the metric. The sample diversity measure needs

---

[2]Code at: https://github.com/martinarjovsky/WassersteinGAN/blob/master/models/dcgan.py

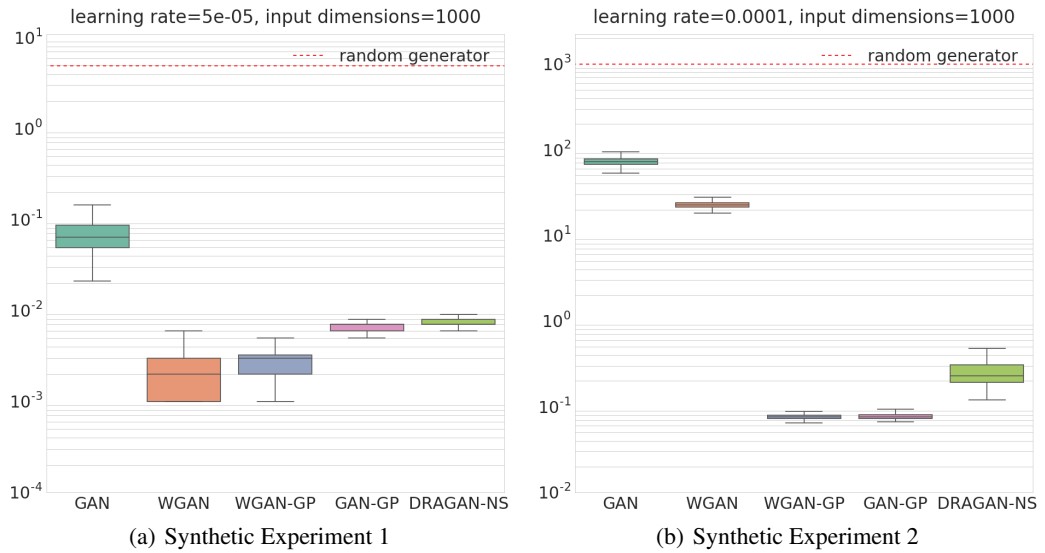

(a) Synthetic Experiment 1    (b) Synthetic Experiment 2

Figure 5: **The square Fréchet distance** between the learned Gaussian and the true Gaussian distribution. For reference, we also plot the distance obtained by a randomly initialized generator with the same architecture as the trained generators. Results are averaged over 1000 runs. Lower values are better.

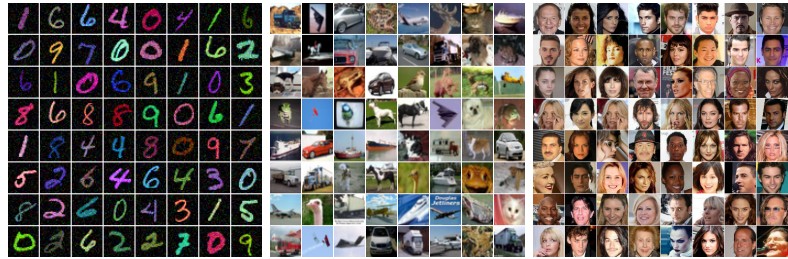

Figure 6: Examples from the three datasets explored in this paper: Color MNIST (left), CIFAR-10 (middle) and CelebA (right).

to be seen in context with the value reported on the test set: too high diversity can mean failure to capture the data distribution. For all other metrics, higher is better.

## 5.2 VISUAL SAMPLE INSPECTION

By visually inspecting the results of our models, we noticed that applying gradient penalties to the non-saturating GAN results in more stable training across the board. When training the non-saturating GAN with no gradient penalty, we did observe cases of severe mode collapse (see Figure 17). Gradient penalties improves upon that, but we can still observe mode collapse. Each non-saturating GAN variant with gradient penalty (DRAGAN-NS and GAN-GP) only produced mode collapse on one dataset, see Figure 21). We also noticed that for certain learning rates, WGAN-GPs fail to learn the data distribution (Figure 22). For the GAN-GP and DRAGAN-NS models, most hyperparameters produced samples of equal quality - the models are quite robust. We show samples from the GAN-GP, DRAGAN-NS and WGAN-GP models in Figures 18, 19 and 20.

## 5.3 METRICS

We show that gradient penalties make non-saturating GANs more robust to hyperparameter changes. For this, we report not only the best obtained results, but rather a box plot of the obtained results showing the quartiles obtained by each sweep, along with the top 10 best results explicitly shown in the graph (note that for each model we tried 27 different hyperparameter settings, corresponding to 3 discriminator learning rates, 3 generator learning rates and 3 generator input sizes). We report two Inception Score metrics for CIFAR-10, one using the standard Inception network used when the metric was introduced (Salimans et al., 2016), trained on the Imagenet dataset, as well as a VGG style network trained on CIFAR-10 (for details on the architecture, we refer the reader to Rosca et al. (2017)). We report the former to be compatible with existing literature, and the latter to obtain a more meaningful metric, since the network doing the evaluation was trained on the same dataset as the one we evaluate, hence the learned features will be more relevant for the task at hand. When reporting sample diversity, we subtract the average pairwise image similarity (as reported by MS-SSIM) computed as the mean of the similarity of every pair of images from 5 batches from the test set. Note that we can only apply this measure to CelebA, since for datasets such as CIFAR-10 different classes are represented by very different images, making this metric meaningless across class borders. Since our models are completely unsupervised, we do not compute the similarity across samples of the same class as in (Odena et al., 2016). The Inception Score and sample diversity metric results can be seen in Figure 8. The results obtained using the Independent Wasserstein critic on all datasets can be found in Figure 7.

## 5.4 KEY TAKEAWAYS FROM REAL DATA EXPERIMENTS

When analyzing the results obtained by training non-saturating GANs using gradient penalties (GAN-GP and DRAGAN-NS), we notice that:

- Both gradient penalties help when training non-saturating GANs, by making the models more robust to hyperparameters.

- On CelebA, for various hyperparameter settings WGAN-GP fails to learn the data distribution and produces samples that do not look like faces (Figure 22). This results in a higher sample diversity than the reference diversity obtained on the test set, as reported by our diversity metric - see Figure 8(a) which compares sample diversity for the considered models across hyperparameters. The same figure shows that for most hyperparameter values, the WGAN-GP model produces higher diversity than the one obtained on the test set (indicating failure to capture the data distribution), while for most hyperparameters non-saturating GAN variants produce samples with lower diversity than that of the test set (indicating mode collapse). However, WGAN-GP is closer to the reference value for more hyperparameters, compared to the non-saturating GAN variants.

- Even if we are only interested in the best results (without looking across the hyperparameter sweep), we see that the gradient penalties tend to improve results for non-saturating GANs.

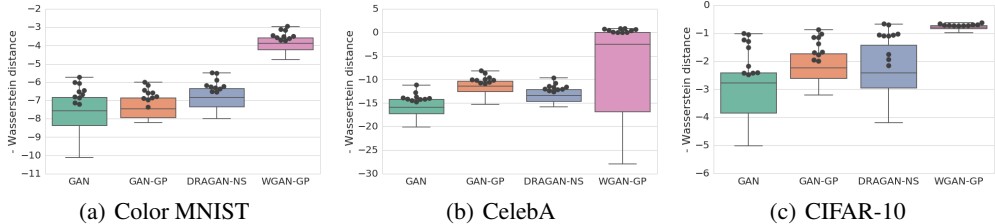

(a) Color MNIST        (b) CelebA        (c) CIFAR-10

Figure 7: Negative Wasserstein distance estimated using an independent Wasserstein critic on the three datasets we evaluate on. The metric captures overfitting to the training data and low quality samples. Higher is better; the 10 black dots represent the results obtained with the 10 best hyperparameter settings.

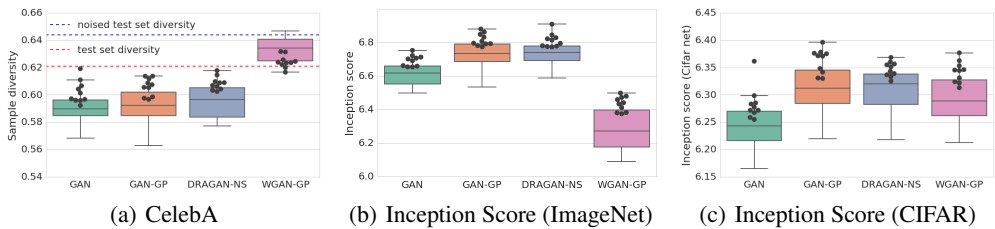

(a) CelebA        (b) Inception Score (ImageNet)        (c) Inception Score (CIFAR)

Figure 8: Left plot shows sample diversity results on CelebA. It is important to look at this measure relative to the measure on the test set: too much diversity can mean failure to capture the data distribution, too little is indicative of mode collapse. To illustrate this, we report the diversity obtained when adding normal noise with zero mean and 0.1 standard deviation to the test set: this results in more diversity than the original data. The black dots report the results closest to the reference values obtained on the test set by each model. Middle plot: Inception Score results on CIFAR-10. Right most plot shows Inception Score computed using a VGG style network trained on CIFAR-10. As a reference benchmark, we also compute these scores using samples from test data split; diversity: 0.621, Inception Score: 11.25, Inception Score (VGG net trained on CIFAR-10): 9.18.

- The non-saturating GAN trained with gradient penalties produces better samples which give better Inception Scores, both when looking at the results obtained from the best set of hyperparameters and when looking at the entire sweep.

- While the non-saturating GAN variants are much faster to train than the WGAN-GP model (since we do only one discriminator update per generator update), they perform similarly to the WGAN-GP model. Thus, non-saturating GANs with penalties offer a better computation versus performance tradeoff. When we trained WGAN-GP models in which we update the discriminator only once per generator update, we noticed a decrease in sample quality for all datasets, reflected by our reported metrics, as seen in Figure 15.

- When looking at the independent Wasserstein critic results, we see that the WGAN-GP models perform best on Color MNIST and CIFAR-10. However, on CelebA the Independent Wasserstein Critic can distinguish between validation data examples and samples from the model (see Figure 7(b)). This is consistent with what we have seen by examining samples: the hyperparameters which result in samples of reduced quality are the same with a reduced negative Wasserstein distance.

- The sample diversity metric and the Independent Wasserstein critic detect mode collapse. When DRAGAN-NS collapses for two hyperparameter settings, the negative Wasserstein distance reported by the critic for these jobs is low, showing that the critic captures the difference in distributions, and the sample diversity reported for those settings is greatly reduced (Figure 16).

## 6    DISCUSSION

We have shown that viewing the training dynamics of GANs through the lens of the underlying divergence at optimality can be misleading. On low-dimensional synthetic problems, we showed that non-saturating GANs are able to learn the true data distribution where Jensen-Shannon divergence minimization would fail. We also showed that gradient penalty regularizers help improve the training dynamics and robustness of non-saturating GANs. It is worth noting that one of the gradient penalty regularizers was originally proposed for Wasserstein GANs, motivated by properties of the Wasserstein distance; evaluating non-saturating GANs with similar gradient penalty regularizers helps disentangle the improvements arising from optimizing a different divergence (or distance) and the improvements from better training dynamics.

**Comparison between explored gradient penalties:** As described in Section 2.3, we have evaluated two gradient penalties on non-saturating GANs. We now turn our attention to the distinction between the two gradient penalties. We have already noted that for a few hyperparameter settings, DRAGAN-NS produced samples with mode collapse, while the GAN-GP model did not. By looking at the resulting metrics, we note that there is no clear winner between the two types of gradient penalties. To assess whether the two penalties have a different regularization effect, we also tried applying both (with a gradient penalty coefficient of 10 for both, or of 5 for both), but that did not result in better models. This could be because the two penalties have a very similar effect, or due to optimization considerations (they might conflict with each other).

**Other gradient penalties:** Besides the gradient penalties explored in this work, several other regularizers have been proposed for stabilizing GAN training. Roth et al. (2017) proposed a gradient penalty aiming to smooth the discriminator of $f$-GANs (including the minimax GAN), which we refer to as $f$-GAN-GP, inspired by Sønderby et al. (2016) and Arjovsky & Bottou (2017). Their gradient penalty is different from the ones explored here; specifically, their gradient penalty is weighted by the square of the discriminator's probability of real for each data instance and the penalty is applied to data and samples (no noise is added). In Fisher-GAN (Mroueh & Sercu, 2017), an equality constraint that is added on the magnitude of the output of the discriminator on data as well as samples is directly penalized, as opposed to the magnitude of the discriminator gradients, as in WGAN-GP. Similar to WGAN-GP, the penalty was introduced in the framework of integral probability metrics, but it can be directly applied to other approaches to GAN training. Unlike WGAN-GP, Fisher GAN uses augmented Lagrangians to impose the equality constraint, instead of a penalty method. To the best of our knowledge, this has not been tried yet and we leave it for future work.

The regularizers assessed in this work (the penalties proposed by DRAGAN and WGAN-GP), as well as others (such as $f$-GAN-GP and Fisher-GAN) are similar in spirit, but have been proposed from distinct theoretical considerations. Future study of GAN regularizers will determine how these regularizers interact, and help us understand the mechanism by which they stabilize GAN training and motivate new approaches.

ACKNOWLEDGEMENTS

We thank Ivo Danihelka and Jascha Sohl-Dickstein for helpful feedback and discussions.

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

# A  RESULTS

## A.1  SYNTHETIC EXPERIMENTS

We present here more detailed results for our synthetic experiments.

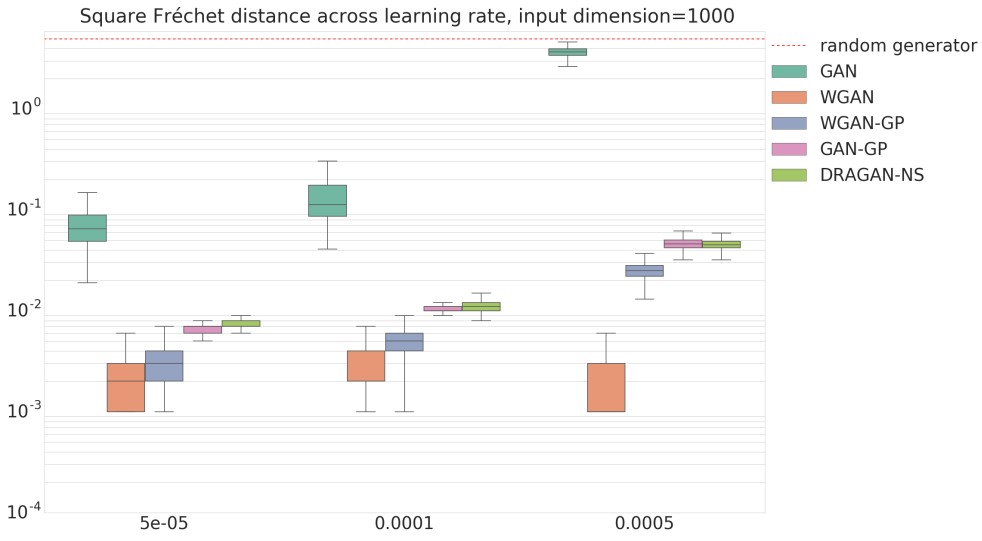

Figure 9: **Synthetic Experiment 1.** The square Fréchet distance between the generated Gaussian parameters and true Gaussian parameters for different GAN variants, when varying the learning rate while keeping the input dimension fixed. Results averaged over 1000 runs. Lower values are better.

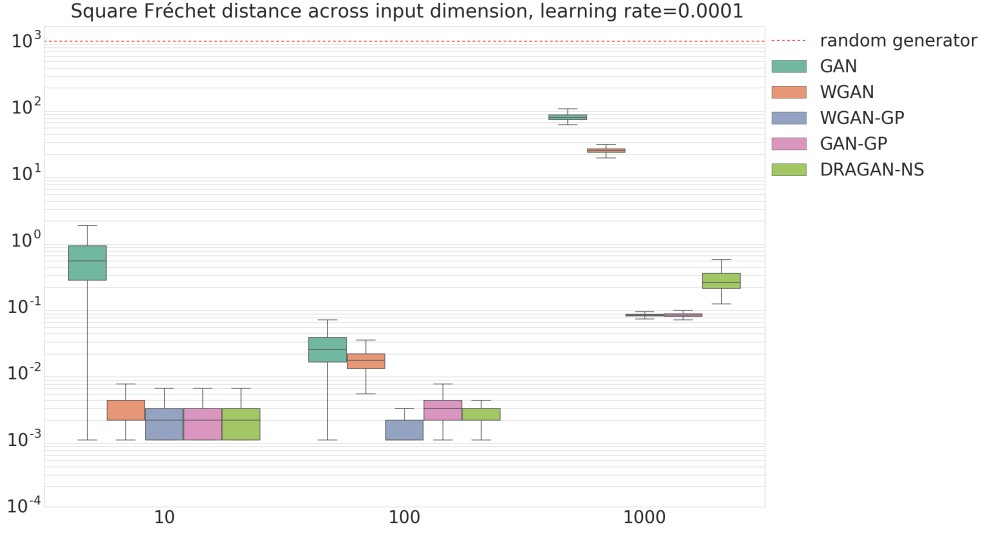

Figure 10: **Synthetic Experiment 2.** The square Fréchet distance between the generated Gaussian parameters and true Gaussian parameters for different GAN variants, when varying the learning rate while keeping the input dimension fixed. Results averaged over 1000 runs. Lower values are better.

## A.2  THE EFFECT OF THE NUMBER OF DISCRIMINATOR UPDATES ON GAN AND GAN-GP

In this section we assess the affects of varying the discriminator update count per generator update. We notice that using 100 discriminator updates per generator update results in a bad distribution fit for the non saturating GAN. GAN-GP scales better with the number of discriminator updates but

increasing the number of discriminator updates does not always result in a closer match to the true distribution for this model either.

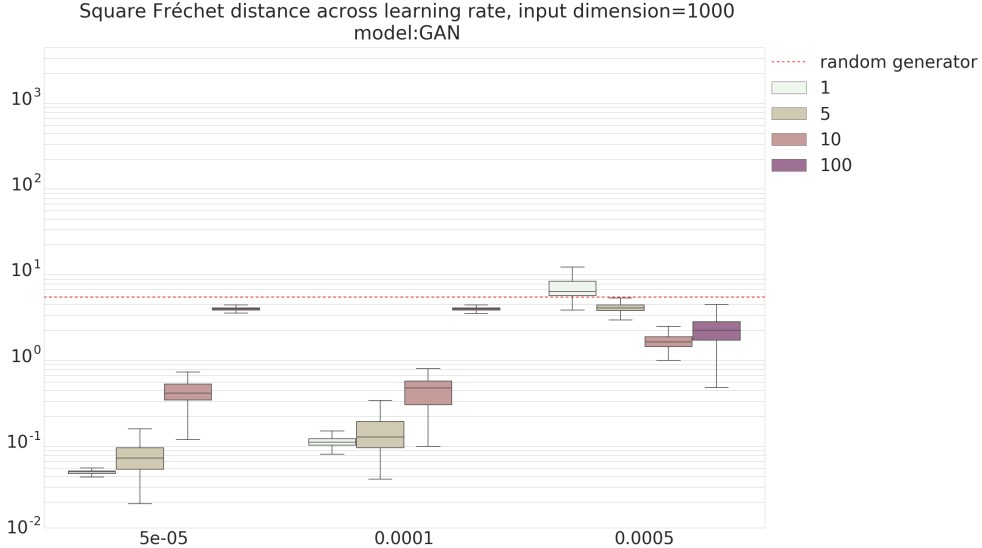

Figure 11: **Synthetic Experiment 1.** The square Fréchet distance between the generated Gaussian parameters and true Gaussian parameters for different number of discriminator updates when training non saturating GANs, with varying the learning rates. Results averaged over 1000 runs. Lower values are better.

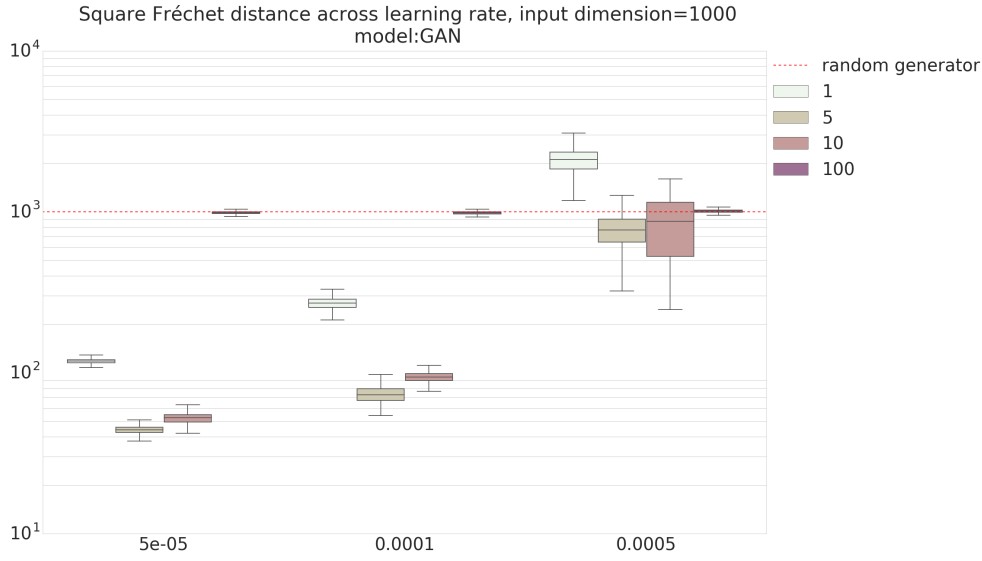

Figure 12: **Synthetic Experiment 2.** The square Fréchet distance between the generated Gaussian parameters and true Gaussian parameters for different number of discriminator updates when training non saturating GANs, with varying the learning rates. Results averaged over 1000 runs. Lower values are better.

## A.3    REAL DATA EXPERIMENTS

We present here generated samples and other evaluation metrics on real data.

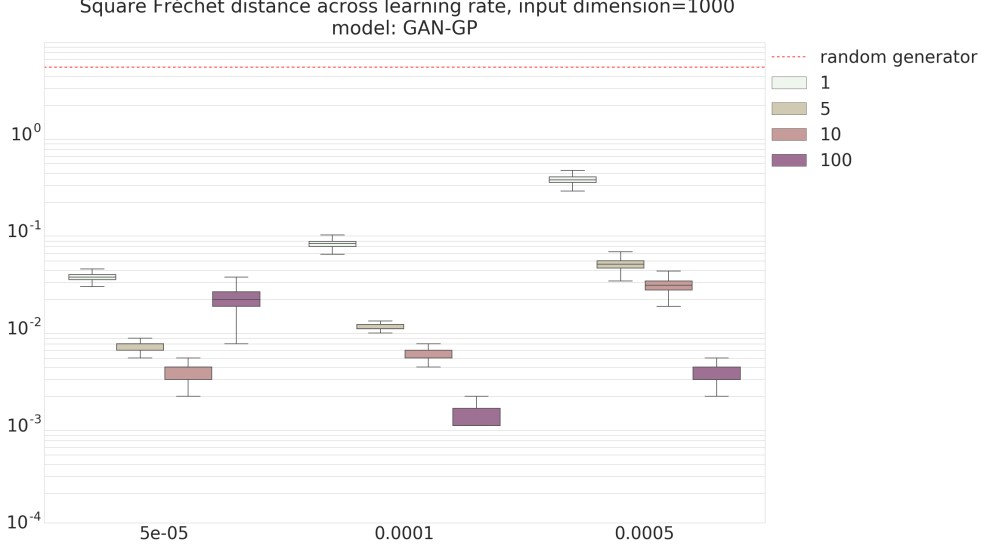

Figure 13: **Synthetic Experiment 1.** The square Fréchet distance between the generated Gaussian parameters and true Gaussian parameters for different number of discriminator updates when training GAN-GP, with varying the learning rates. Results averaged over 1000 runs. Lower values are better.

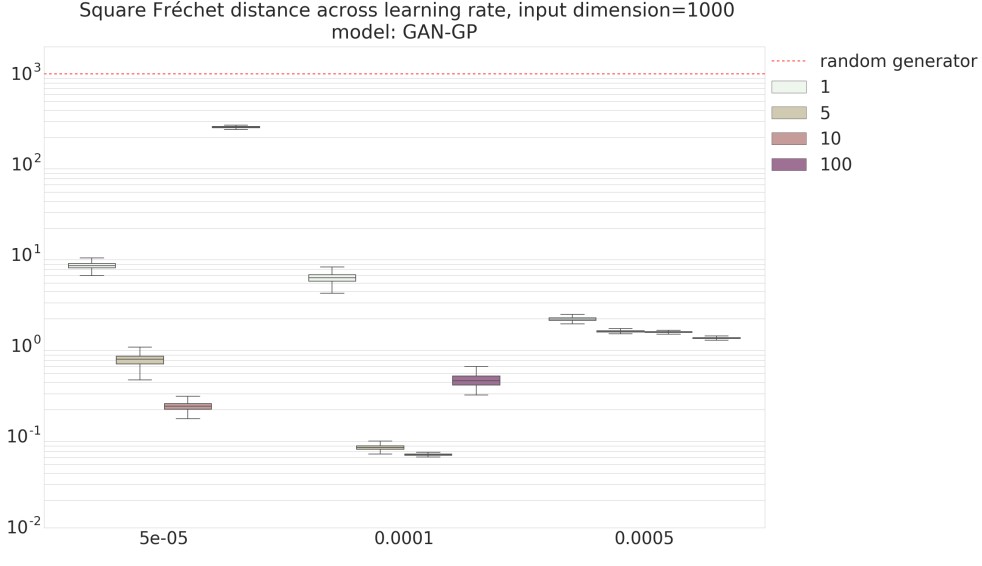

Figure 14: **Synthetic Experiment 2.** The square Fréchet distance between the generated Gaussian parameters and true Gaussian parameters for different number of discriminator updates when training GAN-GP, with varying the learning rates. Results averaged over 1000 runs. Lower values are better.

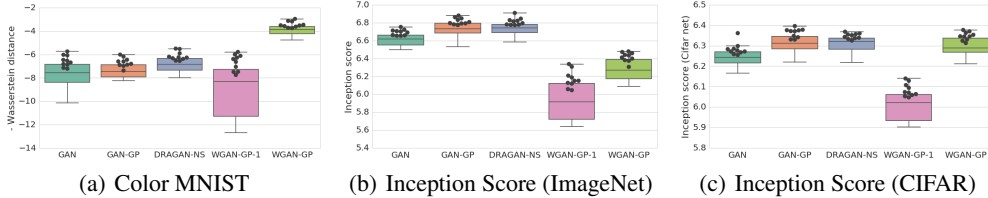

(a) Color MNIST      (b) Inception Score (ImageNet)      (c) Inception Score (CIFAR)

Figure 15: Comparison across models when doing one update for the discriminator in Wasserstein GAN (WGAN-GP-1). The reduced performance in consistent with the observed decrease in sample quality when examining results. Inception Score results obtained on the test set: with Imagenet trained classifier: 11.25, With CIFAR-10 trained classifier: 9.18. Higher is better; the 10 black dots represent the results obtained with the 10 best hyperparameter settings.

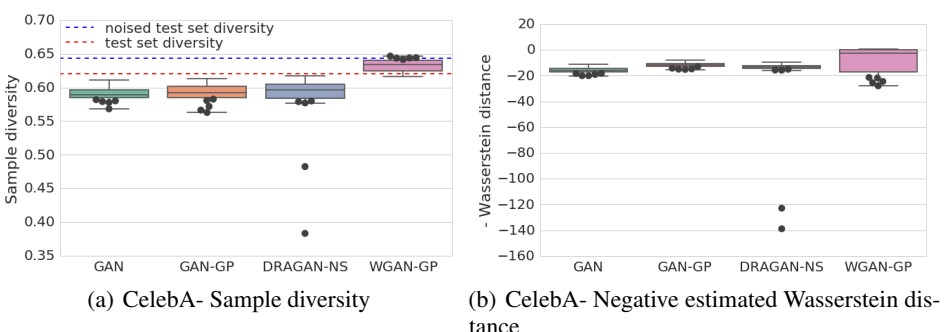

(a) CelebA- Sample diversity      (b) CelebA- Negative estimated Wasserstein distance

Figure 16: The metrics employed are able to capture mode collapse. Looking at the 5 worst values (the black dots) in a hyperparameter sweep according to sample diversity and negative Wasserstein distance as estimated by an Independent Wasserstein critic, we see that these metrics are able to capture the two examples of model collapse that we have seen when training DRGAN-NS on CelebA, as shown in Figure 21. For sample diversity, the worst results are computed by the biggest absolute difference to the reference point (test set diversity), while for negative Wasserstein distance the worst results are computed by choosing the lowest value.

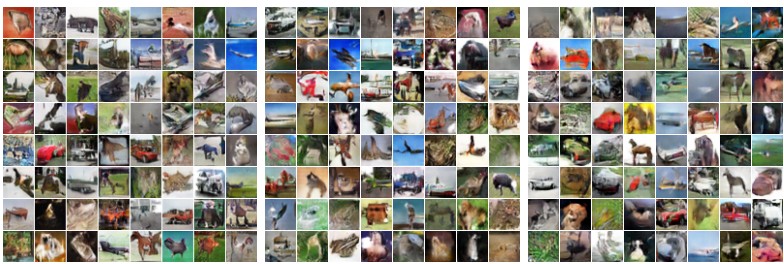

Figure 17: Examples of mode collapse obtained for some hyperparameter settings with non-saturating GAN.

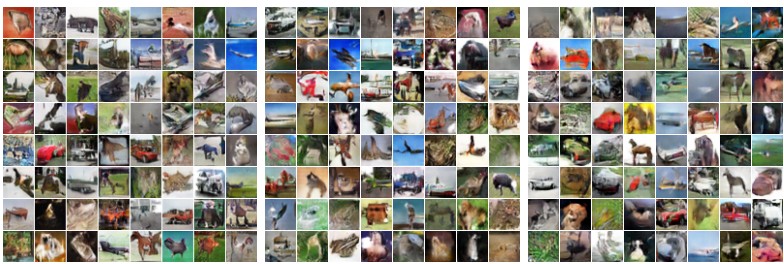

Figure 18: CIFAR-10 samples obtained from the GAN-GP, DRAGAN-NS, and WGAN-GP models.

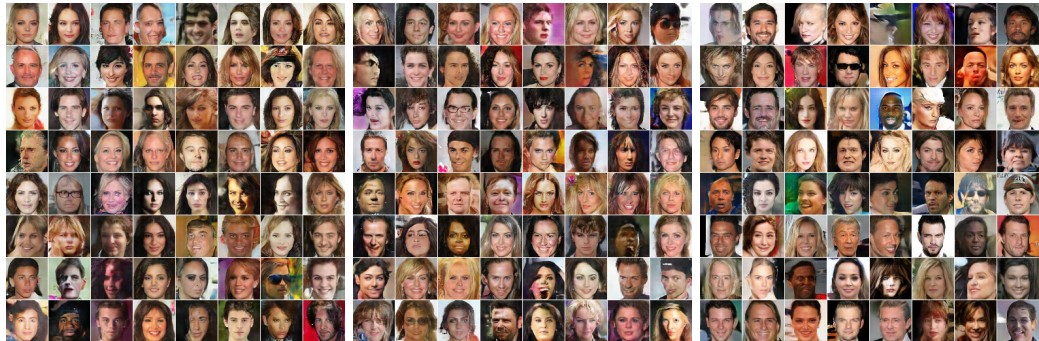

Figure 19: CelebA samples obtained from the GAN-GP, DRAGAN-NS, and WGAN-GP models.

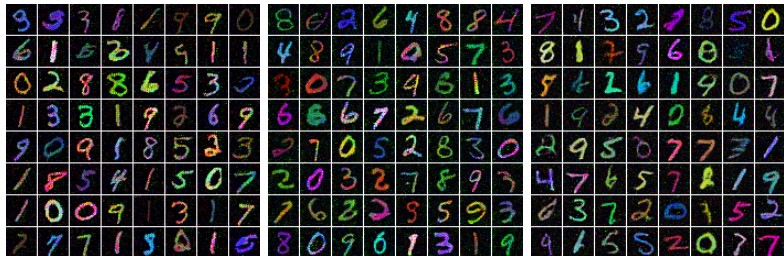

Figure 20: CMNIST samples obtained from the GAN-GP, DRAGAN-NS, and WGAN-GP models.

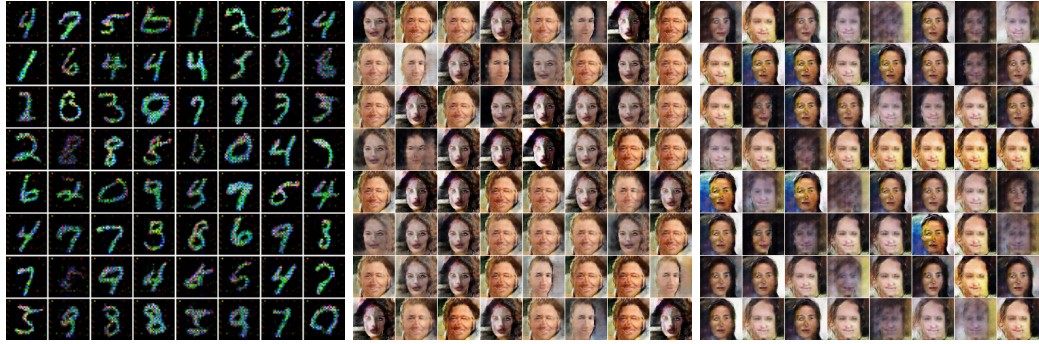

Figure 21: Mode collapse when adding gradient penalties to non-saturating GANs. GAN-GP only had two instances of mode collapse, namely color mode collapse on Color-MNIST (left), while DRAGAN-NS only had two instances of mode collapse, which ocurred when trained on CelebA (right and middle).

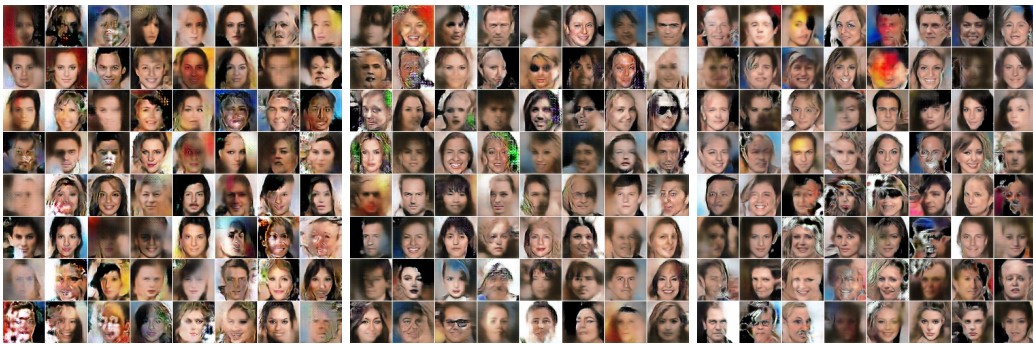

Figure 22: Examples of failure to capture the data distribution with WGAN-GP. The model puts too much mass around the data distribution when trained on the CelebA dataset.

