# OpenReview forum: "Many Paths to Equilibrium: GANs Do Not Need to Decrease a Divergence At Every Step"
_ICLR.cc/2018/Conference — Accept (Poster)_

### Official Review · AnonReviewer3 · 2017-11-24
**Well-written experimental study; light on theory; poses new questions and aims to answer some**

**Rating:** 8
**Confidence:** 5

**Review:**

The submission describes an empirical study regarding the training performance
of GANs; more specifically, it aims to present empirical evidence that the
theory of divergence minimization is more a tool to understand the outcome of
training (i.e. Nash equillibrium) than a necessary condition to be enforce
during training itself.

The work focuses on studying "non-saturating" GANs, using the modified generator
objective function proposed by Goodfellow et al. in their seminal GAN paper, and
aims to show increased capabilities of this variant, compared to the "standard"
minimax formulation. Since most theory around divergence minimization is based
on the unmodified loss function for generator G, the experiments carried out in
the submission might yield somewhat surprising results compared the theory.

If I may summarize the key takeaways from Sections 5.4 and 6, they are:
- GAN training remains difficult and good results are not guaranteed (2nd bullet
  point);
- Gradient penalties work in all settings, but why is not completely clear;
- NS-GANs + GPs seems to be best sample-generating combination, and faster than
  WGAN-GP.
- Some of the used metrics can detect mode collapse.

The submission's (counter-)claims are served by example (cf. Figure 2, or Figure
3 description, last sentence), and mostly relate to statements made in the WGAN
paper (Arjovsky et al., 2017).

As a purely empirical study, it poses more new and open questions on GAN
optimization than it is able to answer; providing theoretical answers is
deferred to future studies. This is not necessarily a bad thing, since the
extensive experiments (both "toy" and "real") are well-designed, convincing and
comprehensible. Novel combinations of GAN formulations (non-saturating with
gradient penalties) are evaluated to disentangle the effects of formulation
changes.

Overall, this work is providing useful experimental insights, clearly motivating
further study.

---

> ### Author Response · Authors · 2017-12-16
> **RE: Well-written experimental study; light on theory; poses new questions and aims to answer some**
>
> Thanks for the detailed and thorough review!
>
> We have now updated the paper with a practical considerations section as well as updated the conclusion to reflect some of your take aways, such as:
>
> - GAN training remains difficult and good results are not guaranteed;
> - Gradient penalties work in all settings, but why is not completely clear;
> - NS-GANs + GPs seems to be best sample-generating combination, and faster than WGAN-GP.
> - Some of the used metrics can detect mode collapse.

---

### Official Review · AnonReviewer1 · 2017-11-27
**A great first study**

**Rating:** 4
**Confidence:** 4

**Review:**

Quality: The authors study non-saturating GANs and the effect of two penalized gradient approaches. The authors consider a number of thought experiments to demonstrate their observations and validate these on real data experiments.

Clarity: The paper is well-written and clear. The authors could be more concise when reporting results. I would suggest keeping the main results in the main body and move extended results to an appendix.

Originality: The authors demonstrate experimentally that there is a benefit of using non-saturating GANs. More specifically, the provide empirical evidence that they can fit problems where Jensen-Shannon divergence fails. They also show experimentally that penalized gradients stabilize the learning process.

Significance: The problems the authors consider is worth exploring further. The authors describe their finding in the appropriate level of details and demonstrate their findings experimentally. However, publishing this  work is in my opinion premature for the following reasons:

- The authors do not provide further evidence of why non-saturating GANs perform better or under which mathematical conditions (non-saturating) GANs will be able to handle cases where distribution manifolds do not overlap;
- The authors show empirically the positive effect of penalized gradients, but do not provide an explanation grounded in theory;
- The authors do not provide practical recommendations how to set-up GANs and not that these findings did not lead to a bullet-proof recipe to train them.

---

> ### Author Response · Authors · 2017-12-16
> **Re: A great first study**
>
> Thank you for your review. We hope to have addressed most of your concerns below:
>
> * We don't believe that the paper is premature for the following reasons:
>        - Gradient penalties are helpful to stabilize GAN training, regardless of the cost function. This is also supported by
>           another paper (https://arxiv.org/pdf/1705.07215v4.pdf).
>        - Gradient penalties are a cost effective way to improve the performance of a GAN. Compared to Wasserstein
>          GAN, in which one needs to do 5 discriminator updates per generator update, DRAGAN-NS and GAN-GP still do 1
>          discriminator update per generator update.
>       - Using multiple metrics can provide a better overview of how an algorithm is performing, as opposed to just using
>         inception score.
>     We will include an additional section in the paper that includes this discussion.
>
> * Our empirical approach to the paper is not a disregard for the importance of theory, but rather a push for an encompassing theory which is inline with the experimental results in our paper. We prove empirically that the exported regularization techniques work outside their proposed scopes, thus showing that a different theoretical justification is needed.  In addition, we show that a theoretical view of GAN training as divergence minimization is incompatible with empirical results.  Specifically, the NS-GAN through GAN training can converge on data distributions that gradient updates on the underlying equilibrium divergence would not.  We wish to encourage the research community to continue to explore theories compatible with these observations.
>
> * Please also see the takeaways of AnonReviewer3: “As a purely empirical study, it poses more new and open questions on GAN optimization than it is able to answer; providing theoretical answers is deferred to future studies. This is not necessarily a bad thing, since the extensive experiments (both "toy" and "real") are well-designed, convincing and comprehensible."
>
> * To make the paper easier to read, we will move more results to the appendix.
>
> * Regarding the theory of gradient penalties, this is something we do not have a handle on currently. We show here that gradient penalties work better independently of the theoretical justification they were introduced with. Perhaps a future avenue of work would be to see if these gradient penalties are related to work which tries to analyze and stabilize GANs by looking at the properties of the Jacobian of the vector field associated with the game (see https://arxiv.org/pdf/1705.10461.pdf, https://arxiv.org/abs/1706.04156)
>
> * To clarify when NS-GAN will not work, we will perform experiments which change the number of updates in the discriminator, and see how that affects performance of model. We note however that for the toy data experiments (Section 4) we performed 5 discriminator updates per generator update.

---

> > ### Author Response · Authors · 2018-01-13
> > **Practical Implications and Theoretical Context**
> >
> > Thanks again for your review!  Before the rebuttal process concludes, do you have any outstanding questions regarding our revision? We ensure this includes specific practical suggestions for GAN-training to guide the community.  In regards to your point about theoretical results, we hope our paper serves to encourage future theoretical research compatible with our observed empirical results. We believe this paper tests a prevailing theoretical understanding of GAN training as directly as possible and that these observations may help validate or invalidate later theoretical models.

---

### Official Review · AnonReviewer2 · 2017-11-28
**Good point**

**Rating:** 7
**Confidence:** 3

**Review:**

This paper answers recent critiques about ``standard GAN'' that were recently formulated to motivate variants based on other losses, in particular using ideas from optimal transport.  It makes main points
1) ``standard GAN'' is an ill-defined term that may refer to two different learning criteria, with different properties
2) though the non-saturating variant (see Eq. 3) of ``standard GAN'' may converge towards a minimum of the Jensen-Shannon divergence, it does not mean that the minimization process follows gradients of the Jensen-Shannon divergence (and conversely, following gradient paths of the Jensen-Shannon divergence may not converge towards a minimum, but this was rather the point of the previous critiques about ``standard GAN'').
3) the penalization strategies introduced for ``non-standard GAN'' with specific motivations, may also apply successfully to the ``standard GAN'', improving robustness, thereby helping to set hyperparameters.
Note that item 2) is relevant in many other setups in the deep learning framework and is often overlooked.

Overall, I believe that the paper provides enough material to substantiate these claims, even if the message could be better delivered. In particular, the writing is sometimes ambiguous (e.g. in Section 2.3, the reader who did not follow the recent developments on the subject on arXiv will have difficulties to rebuild the cross-references between authors, acronyms and formulae). The answers to the critiques referenced in the
 paper are convincing, though I must admit that I don't know how crucial it is to answer these critics, since it is difficult to assess wether they reached or will reach a large audience.

Details:
- p. 4 please do not qualify KL as a distance metric
- Section 4.3: "Every GAN variant was trained for 200000 iterations, and 5 discriminator updates were done for each generator update" is ambiguous: what is exactly meant by "iteration" (and sometimes step elsewhere)?
- Section 4.3: the performance measure is not relevant regarding distributions. The l2 distance is somewhat OK for means, but it makes little sense for covariance matrices.

---

> ### Author Response · Authors · 2017-12-16
> **Re: Good point**
>
> Thank you for the review, your comments made the paper more accessible and improves our experiment evaluations on toy data.
>
> * We will replace the l2 distance between the covariance matrices with the Frechet Distance between two Gaussians as used in Heusel et al. (2017) and update our figures accordingly.
> * We will clarify the statement regarding the KL, together with the difference between step and iteration.
> * We will update section 2.3 to ensure that it is more accessible to a wider audience.

---

> ### Author Response · Authors · 2018-01-11
> **Changes in metrics for the synthetic experiments - using Frechet distance**
>
> All the figures for synthetic experiments are now updated to use the Frechet distance between Gaussians, instead of l2 distance. Thank you for your suggestion!

---

### Public Comment · (anonymous) · 2017-11-01
**How to relate the finding of the paper with the results in [Arjovsky 2017]**

 In the paper of "TOWARDS PRINCIPLED METHODS FOR TRAINING GENERATIVE ADVERSARIAL NETWORKS" by Arjovsky, they show two results:
 1. Lemma 1 shows that if the dimension of Z is less than the dimension of X, then g(Z) will be a set of measure 0 in X. This implies that it is almost impossible to generate samples that are similar to true data samples.
 2. Theorem 2.6 shows that with the non-saturating loss function for the generator, the gradient is of generator has infinite expectation and variance. It implies that using non-saturating loss function is not stable.

 In the paper, the authors show that the non-saturating GAN can learn a high dimensional distribution even though the noise is 1-D. This finding seems to be not aligned with the arguments in [Arjovsky 2017]. I would appreciate if the authors could give more intuitive ideas to explain the relation between the experiment results and the theoretical arguments in Arjovsky 2017. Thanks!

---

> ### Author Response · Authors · 2017-11-09
> **Relating Findings to Earlier Results of Arjovsky et al. (2017)**
>
> Thanks for you comments, Xu.  In response,
>
> 1.  Lemma 1 does indeed show that g(z) will be a set of measure 0 in X for dim(Z) < dim(X), however, this does not necessarily imply that it’s impossible to generate samples matching the data manifold.  The authors are simply stating that it is plausible that the manifold that the data lies on and the manifold of points produced by the generator are disjoint in X.  This would imply a perfect discriminator may exist between the manifolds.  Further, if one tried to bring these manifolds together by minimizing a JS-divergence, the gradients would be meaningless. This motivated the authors' later development of a softer distance measure and the Wasserstein GAN.
> 2.  Theorem 2.6 assumes that the noise of D and the gradient of D are decorrelated, which may be too strong of an assumption.  The authors acknowledge this and then show empirical gradient norms while training DCGAN, which grow with training iterations.  However, in practice, one typically does not train the Discriminator for so many iterations and thus one may avoid the extreme variance cautioned with this theorem.

---

### Public Comment · ~Leon_Boellmann1 · 2017-11-08
**Correctness of regularization**

Dear authors,

We did a simple exercise is to generate a [-1,1] uniform distribution from a Gaussian distribution using GAN with DRAGAN regularization. However, it does not work. What we observed is that D(x) converges to a function with a hump and therefore all the generated samples are concentrated on a small region, instead of uniform distribution. We adopt a sample code from github. The generator has 2 layers and the discriminator has 1 layer.  The lambda is 10.

The reason is that the regularization term pushes the function to have some slope at the data support, which results in the hump shape. Therefore, the generated samples are mostly concentrated in the region with large D(x).

We see that some regularizations make sense mathematically:
- The gradient norm penalty makes sense for WGAN, because the authors in the paper show that "The optimal critic has unit gradient norm almost everywhere under Pr and Pg".
- The application of DRAGAN regularization to WGAN also makes sense because it shows in the paper that there is minor difference between the unit norm argument and the actual application of WGAN-GP, therefore it only applies the gradient penalty in the neighborhood of the data samples. The same holds for the paper of "On the regularization of Wasserstein GANs".
- The regularization term in the paper of "Stabilizing Training of Generative Adversarial Networks through Regularization" makes sense because by Taylor expansion, the noise perturbation at the input approximately adds a regularization term at the objective function. And the training with noise is justified theoretically in the paper of "Towards principled methods for training generative adversarial networks".

However, the application of DRAGAN regularization to the original GAN needs justification. For the original GAN, the optimal D(x) is 1/2 on the data support and hence its gradient is zero. The DRAGAN regularization, however, pushes the gradient norm to 1, which makes the training converge to a wrong value. If we know the regularization is fundamentally and mathematically wrong, why do we investigate its performance?

---

> ### Author Response · Authors · 2017-11-11
> **DRAGAN Regularization Observations**
>
> Thanks for the comment, Leon.
>
> As a quick validation of your architectures and your training code, did you first confirm that you were able to fit the [-1,1] uniform distribution with standard GAN or with improved WGAN?  Also, when you say that your discriminator has one layer, I’m assuming you mean it has one hidden layer and is capable of producing non-linear decision boundaries? An affine discriminator would be insufficient.
>
> To your point about the theoretical correctness of the penalty, the v1 DRAGAN paper (https://arxiv.org/pdf/1705.07215v1.pdf) “How to Train Your DRAGAN” first introduces this regularization penalty onto the *original* GAN discriminator objective (defined as the minimax GAN variant in our paper) as seen in Algorithm 1.  However, this paper actually had an error that their noisy data was not even centered on the original data manifold!  Despite this bug, DRAGAN still succeeded in producing better samples.
>
> The regularization is not necessarily 'fundamentally wrong'. Instead, it is very counterintuitive that it works, given our current level of theoretical understanding. That means that the empirical results showing that it works are more interesting. Empirical results are mostly useful to science when they are surprising. If we experimented with a method that theory predicts should work and it worked, we would not have learned anything. Our results are surprising because we experimented with a method that the theory does not predict should work and yet it worked. This suggests that the theory is at best incomplete and needs to be revised.
>
> For your particular experimental issue, the regularization should be applied in a region *around* the real-data manifold, not over the entire real-data region.  If data manifold is 1D, you should not be applying the DRAGAN penalty throughout the entire region of [-1,1], only at the boundaries.  In higher dimensions, these perturbations will almost always be off-manifold.

---

> > ### Public Comment · ~Leon_Boellmann1 · 2017-11-12
> > **Observation**
> >
> >  Dear authors,
> >  I was using the code shared on the github link in the DRAGAN paper (https://arxiv.org/pdf/1705.07215v1.pdf). Sorry, there was a typo in my last message. Both the discriminator and the generator have 2 layers. The conventional GAN and WGAN works even without regularization.
> >
> >  I strongly agree with the authors that it is very interesting to investigate something that empirically works but is not fully known in theory. I think that is why deep learning is so attractive to so many people including myself. On the other hand, I also think we should investigate something that makes sense. Decades ago, we already established the "universal approximation" theorem for neural networks. We know that it can fundamentally fits any continuous function on a compact set. If we know it cannot fit certain functions, say a purely linear network, we would not even start training it to fit high-dimensional complicated functions.
> >
> > I think my argument is based on this philosophy. We know that the DRAGAN regularization is actually wrong for the original GAN in some cases, because gradient norm should be 0 at the optimal point (for example the uniform [-1,1] example). It may have good results for some applications in training images, however, for some applications (say, autonomous driving) we cannot take any risk in applying an algorithm that does not work in come corner cases. In this case, I would devote more time to investigate other methods that are fundamentally correct, for example DRAGAN regularization on WGAN.

---

> > > ### Author Response · Authors · 2017-11-13
> > > **Applying DRAGAN Gradient Penalty in Your Example**
> > >
> > > Thanks for the clarification.
> > >
> > > And yes, but just to again reiterate, we are not suggesting that you apply the DRAGAN gradient penalty inside the 1D uniform [-1, 1] region.  Enforcing the gradient norm to be 1 inside here would fail as you described.  You should have no issue fitting this training distribution if you only apply the DRAGAN gradient penalty on the *boundaries* of the data distribution, i.e. -1 + delta^i and +1 + delta^j.

---

### Author Response · Authors · 2018-01-25
**Communicating directly to AC: openreview bugs are hampering our communication**

Yesterday, some of the authors got an e-mail asking us to respond ASAP to a comment from the AC. When we visited the webpage, we could not see a comment, even when logged in. We had earlier posted a reply to a similar comment

We contacted the ICLR program chairs yesterday and were told not to worry about it, that the AC had seen our reply in the meantime.

Today we got another message saying to reply to the AC asap, quoting a comment that we're not able to see.

I contacted the ICLR program chairs a second time and they say that they can't see the messages from the AC to me in the openreview system and that this might be a phishing attempt.

As a further comment, the messages from the AC did not reach all authors. Specifically, they did not reach the first author, who uploaded the submission, but instead went to a collaborator in a different country and time zone.

If the AC really has been trying to get in touch with us, I want to make it clear that we were trying to respond as quickly as possible, but we've been hampered by openreview bugs and miscommunications with the program chairs.

Since the e-mail we received today actually fully quotes the comment that we're not able to see, we actually are able to respond. The comment asks for us to provide more experiments and discussions promised in the rebuttal. We actually did reply to a similar comment earlier explaining the situation, but apparently the AC can't see our reply, presumably due to an openreview bug. Our reply is: the 1st author will upload the latest revision today. It will include the requested discussions, but one of the requested experiments is still running. The experiments are computationally expensive and can't be accelerated without reducing their accuracy. The experiment will definitely be ready for the final copy deadline.

---

> ### Comment · Area_Chair · 2018-01-25
> **These are old emails.**
>
> Bugs were fixed yesterday and a flood of old emails were sent. But rest assured, I've been reading your comments actively. Thanks for your message.

---

### Decision · Program_Chairs · 2018-01-29
**ICLR 2018 Conference Acceptance Decision**

**Decision:**

Accept (Poster)

**Comment:**

AnonReviewers 2 and AnonReviewer 3 rated the paper highly, with AR3 even upgrading their score.  AnonReviewer1 was less generous:

" Overall, it is a good empirical study, raising a healthy set of questions. In this regard, the paper is worth accepting. However, I am still uncomfortable with the lack of answers and given that the revision does not include the additional discussion and experiments promised in the rebuttal, I will stay with my evaluation."

The authors have promised to produce the discussion and new experiments. Given the nature of both (1: the discussion is already outline in the response and 2: the experiments are straightforward to run), I'm inclined to accept the paper because it represents a solid body of empirical work.